# Peripheral Blood Mononuclear Cells Predict Therapeutic Efficacy of Immunotherapy in NSCLC

**DOI:** 10.3390/cancers14122898

**Published:** 2022-06-12

**Authors:** Jacobo Rogado, Fernando Pozo, Kevin Troule, José Miguel Sánchez-Torres, Nuria Romero-Laorden, Rebeca Mondejar, Olga Donnay, Anabel Ballesteros, Vilma Pacheco-Barcia, Javier Aspa, Fátima Al-Shahrour, Arantzazu Alfranca, Ramon Colomer

**Affiliations:** 1Medical Oncology Department, Hospital Universitario Infanta Leonor, Gran Via del Este 80, 28031 Madrid, Spain; 2Instituto de Investigación Sanitaria del Hospital Universitario la Princesa, 28006 Madrid, Spain; jmiguelst@gmail.com (J.M.S.-T.); nuriaromerolaorden@gmail.com (N.R.-L.); rmondejars@hotmail.com (R.M.); olga_donnay@telefonica.net (O.D.); anabelballes@gmail.com (A.B.); vpbarcia@yahoo.es (V.P.-B.); jaspa@separ.es (J.A.); aalfranca@gmail.com (A.A.); rcolomer@seom.org (R.C.); 3Bioinformatics Unit, Spanish National Cancer Research Centre, 28029 Madrid, Spain; fpozoc@cnio.es (F.P.); ktroule@cnio.es (K.T.); falshahrour@cnio.es (F.A.-S.); 4Medical Oncology Department, Hospital Universitario la Princesa, 28006 Madrid, Spain; 5Department of Medicine and Chair of Personalized Precision Medicine, Universidad Autónoma de Madrid, 28006 Madrid, Spain; 6Medical Oncology Department, Hospital Central de la Defensa Gómez Ulla, 28047 Madrid, Spain; 7Pneumology Department, Hospital Universitario la Princesa, 28006 Madrid, Spain; 8Immunology Department, Hospital Universitario la Princesa, 28006 Madrid, Spain

**Keywords:** non-small cell lung cancer, anti-PD-1 antibodies, immunotherapy, biomarkers, peripheral blood mononuclear cells

## Abstract

**Simple Summary:**

Biomarkers to guide clinical decisions and efficacy are limited in advanced non-small cell lung cancer’s anti-PD-1 immune checkpoint inhibitors. We prospectively explored baseline peripheral blood mononuclear cells in order to asses’ immunotherapy predictors in this setting. We included 39 patients diagnosed with non-small cell lung cancer treated with immunotherapy in the study group and 40 patients with advanced malignancies treated with non-immunotherapy treatment, as control group. We detected that high baseline levels of circulating T cell subpopulations related to tissue lymphocyte recruitment are associated with poorer outcomes of immunotherapy-treated advanced non-small cell lung cancer patients, and these differences were specific to immunotherapy-treated patients.

**Abstract:**

In lung cancer immunotherapy, biomarkers to guide clinical decisions are limited. We now explore whether the detailed immunophenotyping of circulating peripheral blood mononuclear cells (PBMCs) can predict the efficacy of anti-PD-1 immunotherapy in patients with advanced non-small-cell lung cancer (NSCLC). We determined 107 PBMCs subpopulations in a prospective cohort of NSCLC patients before starting single-agent anti-PD-1 immunotherapy (study group), analyzed by flow cytometry. As a control group, we studied patients with advanced malignancies before initiating non-immunotherapy treatment. The frequency of PBMCs was correlated with treatment outcome. Patients were categorized as having either high or low expression for each biomarker, defined as those above the 55th or below the 45th percentile of the overall marker expression within the cohort. In the study group, three subpopulations were associated with significant differences in outcome: high pretreatment levels of circulating CD4+CCR9+, CD4+CCR10+, or CD8+CXCR4+ T cells correlated with poorer overall survival (15.7 vs. 35.9 months, HR 0.16, *p* = 0.003; 22.0 vs. NR months, HR 0.10, *p* = 0.003, and 22.0 vs. NR months, HR 0.29, *p* = 0.02). These differences were specific to immunotherapy-treated patients. High baseline levels of circulating T cell subpopulations related to tissue lymphocyte recruitment are associated with poorer outcomes of immunotherapy-treated advanced NSCLC patients.

## 1. Introduction

The programmed cell death receptor PD-1 plays an essential role in the immune system homeostasis, regulating T-cell responses and immune tolerance [1]. Immune checkpoint inhibitors (ICIs) targeted to PD-1 or its ligand PD-L1 have changed the landscape in non-small cell lung cancer (NSCLC) treatment [2,3]. PD-L1 tumor tissue expression is a limited predictor of anti-PD-1 efficacy [2,3,4]. In some clinical trials, it has been demonstrated that the greater the expression of PD-L1, the greater the response observed. However, in other studies, this association was not observed. In addition, PD-L1 is a dynamic and heterogeneous biomarker. It may be expressed in a different way depending on where the biopsy was taken and at what disease time [2,3,4]. Therefore, we think PD-L1 is not a good reliable biomarker, although at the present time, it is the only one validated. Other tissue biomarkers such as tumor mutational burden (TMB) have been explored, yielding controversial results. Some studies reported that the greater the TMB, the greater the release of tumor neoantigens, thus leading to greater activation of the immune system and producing more benefits in terms of immunotherapy treatments [5]. However, there are other studies that did not correlate these findings, so this is not a reproducible biomarker, either [6]. 

Our previous studies identified patient-dependent predictors of immunotherapy efficacy, such as the development of immune-related side effects or the presence of a high body mass index [7,8]. Regarding immune-related adverse events (IrAEs), we carried out a retrospective study in 2019. We included 106 patients with different tumors, treated with anti-PD-1 inhibitors. We observed how patients who developed IrAEs during treatment presented significantly better outcomes in terms of objective response and survival [7]. However, this is not a basal biomarker that can be detected before the start of treatment, so we cannot select patients at the beginning of the therapies using the development of an IrAE. On the other hand, in another of our previously designed studies in which we knew the proinflammatory state of obese or overweight patients, we wanted to see whether excess weight was associated with a better response to immunotherapy [8]. In this sense, we also saw that overweight patients had a better outcome in terms of objective response and survival. In addition, when these overweight patients developed an IrAE, this beneficial effect on survival and response multiplied [8]. 

Likewise, neutrophil-to-lymphocyte ratio (NLR), neutrophil count percentage (NCP), and derived-NLR combined with lactate dehydrogenase (LDH) levels—LIPI score [9,10]—have been proposed as non-tissue predictors of immunotherapy outcome. However, their role in NSCLC is not clear [4]. In some studies, a favorable LIPI score has been associated with better survival [9]. Regarding NLR or NCP, we verified in another previous study that patients with NSCLC treated with nivolumab who had an NLR below 5 or an NCP below 80% had better outcomes in terms of survival. However, this has not been verified in other work, and in addition, the cut-off points for NLR, dNLR, or NCP vary between the different studies. Therefore, we do not think that they are reproducible biomarkers, either [4]. 

For this reason, we explored whether the use of a detailed immunophenotypic evaluation of peripheral blood mononuclear cell (PBMC) subpopulations could be of value to define predictive biomarkers of immunotherapy efficacy in advanced NSCLC patients.

## 2. Materials and Methods

We analyzed 107 immune populations in peripheral blood (PB) using flow cytometry, including subsets of T and B lymphocytes, and natural killer and myeloid cells using specific antibodies (see Table 1). The study group consisted of patients diagnosed with advanced NSCLC before single-agent nivolumab or pembrolizumab treatment. The control group included patients with advanced cancer before treatment with non-immunotherapy agents. The patients were followed from the start of immunotherapy until data cut-off on 31 July 2019 (median follow-up 5.03 months, range 0–31.34). The frequency of immune subpopulations was correlated with clinical–demographic characteristics and treatment outcome in terms of OS, and PFS was also analyzed. We performed univariate and multivariate analyses.

### 2.1. PBMC Isolation

For PBMC isolation, 10 mL of whole blood was collected in EDTA tubes, and PBMCs were obtained using Ficoll-Paque (Panbiotech, Aidenbach, Germany), following the manufacturer’s instructions. The cells were frozen in fetal bovine serum (HyClone TM) containing 10% DMSO (Unilab) and subsequently maintained at −80 °C, or liquid N_2_, for long-term storage. 

### 2.2. Flow Cytometry

For the flow cytometry assays, PBMCs were thawed using 10 mL of medium RMPI 1640 (GIBCO) supplemented with 10% fetal bovine serum and 5% penicillin–streptomycin (Biowest, Nuaillé, France). Then, the cells were incubated with blocking FcR reagent (Miltenyi, Bergisch Gladbach, Germany) for 15 min at 4 °C, and subsequently with different combinations of monoclonal antibodies, including: anti-human-CD47-FITC, -CCR9-Alexa 488, -CD126-PE, -KIR-PE, -CD3-PerCP, -HLA-DR-PerCP, -CLA-Alexa647, -CD210-Alexa647, -LAG3-Alexa647, -CD244-APC, -CD184-PECy7, -CD4-APCH7, -CD8-APCH7, -CD14-APCH7, -CD11c-V450, -IL17a-V450, -CD123-V510, -CD19-V500, -IFNγ-V500, and -CD8-V500 (BD Becton Dickinson, Franklin Lakes, NJ, USA); anti-human-IL15Ra-FITC, and -Thrombospondin-1-PECy7 (Invitrogen, Waltham, MA, USA); anti-human-CCR10-FITC, and -NKG2A-PerCP (R&D systems, Minneapolis, MN, USA); anti-human-TIM-3-FITC, -Tie2-PE, -β7-APC, and -CTLA-4-PECy7 (Biolegend, San Diego, CA, USA); anti-human-NKG2c-Viobright FITC, -ADAM-8-PE, and -SLAN-PE (Miltenyi); and anti-human-GRK-2 (Santa Cruz, Dallas, TX, USA) for 30 min at 4 °C (see Table 1). Finally, the cells were washed and resuspended in 200 μL of PBS 1×. 

The remaining material was frozen and stored in the biobank of our center, as reflected in the patient information sheet and in the informed consent.

All of the samples were acquired on a BD FACSCanto II flow cytometer (BD Becton Dickinson), and the data obtained were analyzed using FlowJo software (BD Becton Dickinson).

### 2.3. Statistical Methods

For the descriptive statistics, all patients were included. Patients with more than six missing values in the biomarker data were excluded for survival analysis. 

For the continuous variables, means and standard deviations (SDs) are shown, and the median and interquartile range (IQR) are employed for variables that do not follow a normal distribution. Finally, the discrete variables are summarized with their frequencies. The progression-free survival (PFS) data were calculated from the first dose of immunotherapy until progression or death, or censored at the last date of the follow-up in the non-small cell lung cancer cohort, and were calculated from the first dose of non-immunotherapy treatment until progression or death, or censored at the last date of the follow-up in the control group. The overall survival (OS) data were calculated from the advanced cancer diagnosis until death or censored at the last follow-up date.

Survival analysis was performed according to the percentage of expression in the PB cells of the 107 immune populations, both in the group of patients under study (non-small cell lung cancer treated with immunotherapy) and in the control group (non-immunotherapy cohort).

Regarding the cut-off point selection, to perform the subdivision in high expression or low expression of an immune biomarker in PB, we decided to make an intermediate cut-off point in an intermediate value, for a better interpretation and to try to avoid further biases. For this reason, the patients were divided into each biomarker according to whether they had high expression (greater than or equal to the 55th percentile) or low expression (less than or equal to the 45th percentile).

The study was approved by the ethics committee of the Hospital Universitario La Princesa on 22 December 2016 under the code 2918.

## 3. Results

We included 79 patients: 39 in the study group and 40 in the control group. The clinical characteristics are described in Table 2. 

### 3.1. Study Group Population

In the study group, 39 patients diagnosed with NSCLC were included. The median age in this group was 69 years old. Only three women were included (7.69%), which could introduce some bias that we will try to avoid with the multivariate analysis. All of the patients were current or former smokers. Only one patient had history of autoimmune disease (psoriasis). Most of the patients had a good performance status (ECOG PS 0 and ECOG PS 1: over 70%). The most frequent histology was non-squamous cell NSCLC (64.10%). High PD-L1 expression (≥50) was present in 56.41% of the patients. No differences were detected between high PD-L1 expression and the different histology included: 66.7% of the patients were diagnosed with adenosquamous NSCLC (*p* value = 0.7), 48% were diagnosed with non-squamous NSCLC (*p* value = 0.15), and 72.3 were diagnosed with squamous NSCLC (*p* value = 0.19). The most frequent site of metastasis was the central nervous system. Twenty patients received single-agent pembrolizumab and 19 nivolumab. 

### 3.2. Control Group Population

In the control group, 40 patients with different histologies and cancer types were included. The median age in this group was 68 years old. Sixteen women were included (40%). Twenty patients (57.5%) were current or former smokers. Most of the patients had a good performance status (ECOG PS 0 and ECOG PS 1: 68%). Histology and cancer types: 9 patients were diagnosed with non-small cell lung cancer, 2 patients with small-cell lung cancer, 24 patients with pancreatic adenocarcinoma, 4 patients with colon cancer, and 1 patient was diagnosed with breast cancer. The most frequent treatments received: (all received non-immunotherapy treatment): 29 patients received exclusive chemotherapy treatment, 3 patients received tyrosine kinase inhibitors against anaplastic lymphoma kinase, and 1 patient received hormone therapy. The most frequent site of metastasis was the liver (55%). The prognosis of the patients was balanced when compared with the study group, having similar involvement in vital organs, as was the case of the CNS in the study group. However, metastatic involvement at different sites may be a limitation in the interpretation of the results, which we hope to minimize as much as possible with the multivariate analysis as we can show in the survival analysis. The clinical characteristics are described extensively in Table 2. 

### 3.3. Survival Analysis

Three patients were excluded from the study group in the survival analysis, and three from the control group. Three detailed T-cell subsets in the study group demonstrated a consistent and significant survival effect. High baseline levels of T CD4+CCR9+ cells had poorer outcomes for OS (15.7 vs. 35.9 months, HR 0.16, CI 95% 0.05–0.55, *p* = 0.003) and also PFS (2.6 months vs not reached (NR), HR 0.3, CI 95% 0.11–0.82, *p* = 0.019) (Figure 1a); high baseline levels of T CD4+CCR10+ cells only showed poorer outcomes in terms of OS (22.0 months vs. NR, HR 0.10, CI 95% 0.02–0.47, *p* = 0.003) (Figure 1c); and high baseline levels of T CD8+CXCR4+ cells had worse OS (22.0 months vs. NR, HR 0.29 CI 95% 0.10–0.86, *p* = 0.02) and a trend towards significance in PFS (5.0 vs. 14.2 months, HR 0.43, CI 95% 0.15–1.25, *p* = 0.12) (Figure 1e). In the control group, none of these subpopulations exhibited comparable behavior: high baseline levels of PB T cell CD4+CCR9+, CD4+CCR10+, and CD8+CXCR4+ displayed no survival significance (Figure 1b,d,f). The multivariate analysis was adjusted by clinical–demographic characteristics including immune-related adverse event development and excess weight at the start of treatment.

There are no differences between the high or low expressors with the baseline levels of T-helper CCR9+ cells and the objective response rate in the NSCLC study group (odds ratio 0.50, CI 95% 0.13–1.92, *p* = 0.31), but the logistic regression shows a trend towards significance (33.33% response rate in high expressors vs. 50% in low expressors). However, we detected that low baseline levels of this cell subset in peripheral blood produced an increase in survival, regardless of whether the patient developed an objective response or not. In non-responder patients, in this case, we detected that high baseline levels of T CD4+CCR9+ cells had poorer outcomes for OS (10.4 vs. 34.3 months, HR 0.15, CI 95% 0.03–0.71, *p* = 0.017) and PFS (2.1 months vs. 6.6 months, HR 0.09, CI 95% 0.01–0.52, *p* = 0.006, Figure 2). This subset of PBMCs was the only one of all those showing benefit in survival, in which we detected that the benefit in progression-free and overall survival was maintained, even though the patients did not show an objective radiological response. 

In addition to studying the detailed subsets, we also analyzed broader cell sets such as CD4+T cells, CD8+T cells, B cells, monocytes, and dendritic cells. Except for high baseline levels of CD4+ T cells (18.7% in relative count) that showed worse OS, (22.0 vs. 35.9 months; HR 0.29, CI 95% 0.09–0.93, *p* = 0.03), we did not observe a significant association with survival (see Figure 3). The worse OS in these high baseline levels of T-helper cells was not exhibited in the control group. Descriptions of the best immune subpopulation biomarkers in the overall survival setting are shown in Table 3.

Neither cell subset detected as a possible predictive biomarker in NSCLC (study group) was correlated with survival in the control group.

## 4. Discussion

We report the first study showing that the chemokine receptor expression in PBMCs correlates with anticancer immunotherapy outcomes. In our study, the survival of advanced lung cancer treated with immunotherapy was significantly reduced in those patients with high baseline levels of CCR9+ or CCR10+ CD4+ T cells, or CXCR4+ CD8+ T cells. Our findings were specific to immunotherapy, since they were not observed in the control group that included patients with advanced malignancies who received a treatment other than immunotherapy. 

Chemokines and chemokine receptors regulate immune cells and mediate the homing process in the immune system, and it was suggested that the expression levels of receptors and/or their ligands might be relevant in the processes of carcinogenesis and metastasis as well as for cancer immunotherapy [11]. However, only a limited number of reports have studied the relationship between chemokine receptor expression and prognosis in cancer patients. All of these studies differ from ours in that chemokine receptors were not explored in circulating lymphocytes, but rather in either cancer cells or tumor-infiltrating lymphocytes (TILs); that patient’s therapy for advanced cancer, if any, did not include immunotherapy; and that the assessment of prognosis did not include a control group. 

An immunohistochemical retrospective study of 60 gliomas showed that high CCR10 expression, which was observed in about half the tumors, correlated with poor survival [12]. CCR10 and its ligand CCL28 are known to participate in the recruitment of tumor TReg lymphocytes [13], and a decrease in CD4+ CCR10+ cells may reflect a reduced global presence of CCR10-expressing TRegs and, therefore, a reduced recruitment of these cells by the tumor. Such an effect might compromise the generation of an immunosuppressive microenvironment, thus favoring the anticancer effect of immunotherapy. Our study did not characterize whether the CCR10-expressing CD4+ subpopulation in our patients was composed by TRegs, and this will need to be addressed in the future.

A study of 76 lung adenocarcinoma biopsies showed that high levels of CCR9 detected by immunohistochemistry, which occurred in 63% of samples, correlated with worse overall survival. Twenty-four cases were metastatic. The treatment details other than surgery were not reported [14]. Interestingly, additional Transwell experiments showed that CCR9/CCL25 promoted the migration and invasiveness of lung cancer stem cells isolated from A549 cells, and the authors suggested that CCR9 or its ligand CCL25 could be used as a therapeutic target for lung adenocarcinoma. 

In a series of 176 tumor samples from patients with advanced lung cancer, in which CXCR4 was expressed in most tumors, CXCR4 overexpression was associated with significantly poorer survival [15]. The authors suggested that CXCR4 might be a potential therapeutic target.

In another study, CXCL12, the CXCR4 ligand, was expressed in most non-small cell lung cancer tissue sections obtained from stage IA to IIB non-small cell lung cancer patients undergoing operation. The disease recurrence rates in the subgroup of 29 adenocarcinoma patients tended to correlate with high CXCL12 expression in the tumor [16]. The authors also observed that CXCL12 correlated with a higher degree of tumor inflammation and suggested that the concomitant presence of activated TILs CD4+CXCR4+ CD69+ might influence tumor progression by shaping the immune cell population infiltrating lung adenocarcinomas. Since it has been described that CXCR4 blockade may increase the number of tumor-infiltrating T lymphocytes and display synergistic effects with immune checkpoint inhibitors [17], this may help explain the adverse outcome of our immunotherapy-treated patients with high expression of CXCR4 in peripheral T lymphocytes, perhaps related to the creation of a worse scenario for immunotherapy effectiveness.

## 5. Conclusions

In summary, we show that a high expression of CCR9, CCR10, and CXCR4 in the peripheral T lymphocytes of advanced lung cancer patients treated with anti-PD-1 immune checkpoint inhibitors is related to a poorer outcome, and that this effect was not observed in advanced cancer patients not receiving immunotherapy. Immunophenotyping PBMCs may, in the future, provide a novel biomarker approach for selecting cancer patients for immunotherapy, either independently of or complementary to cancer tissue testing. Functional assays will be needed to characterize the PBMC populations that we have described to determine which mechanisms are involved in anti-PD-1-mediated outcomes.

## Figures and Tables

**Figure 1 cancers-14-02898-f001:**
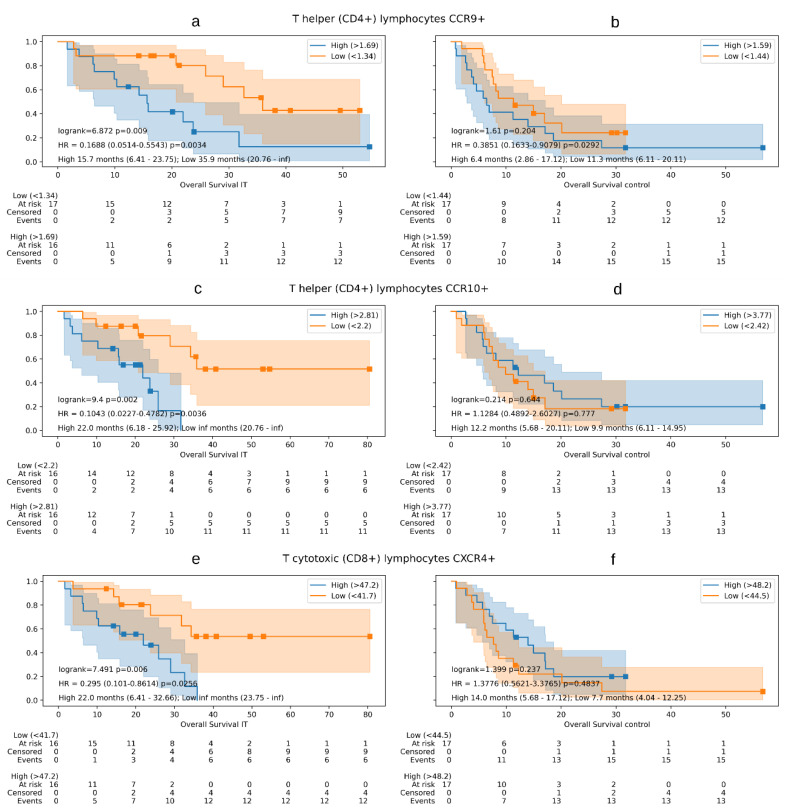
Overall survival Kaplan–Meier curves showing overall survival differences between immunotherapy treatment (IT) study group versus control group. The Kaplan–Meier curves show the differences in survival according the high or low expression of each peripheral blood mononuclear cell subpopulation studied in both groups. In addition, the log-rank test, multivariate cox regression models, and the median overall survival values with their range are reflected. Graphics (**a**,**b**) show overall survival in immunotherapy treatment group (**a**) versus study group (**b**) according to the expression of T-helper lymphocytes CCR9+. Graphics (**c**,**d**) show overall survival in immunotherapy treatment group (**c**) versus study group (**d**) according to the expression of T-helper lymphocytes CCR10+. Graphics (**e**,**f**) show overall survival in immunotherapy treatment group (**e**) versus study group (**f**) according to the expression of T-cytotoxic lymphocytes CXCR4+.

**Figure 2 cancers-14-02898-f002:**
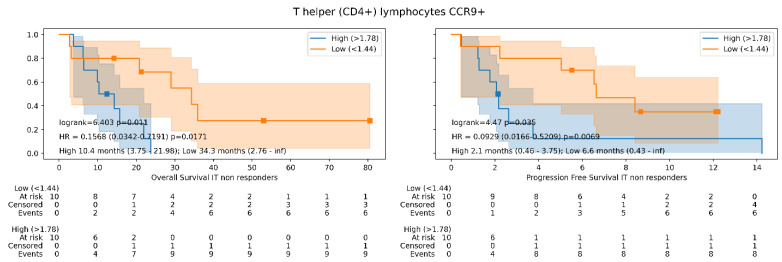
Overall survival and progression-free survival Kaplan–Meier curves showing overall survival differences in study group according to T-helper lymphocytes expressing CCR9 baseline levels in non-responder patients. The Kaplan–Meier curves show the differences in survival according the high or low expression of T-helper lymphocytes expressing CCR9 in the study group. In addition, the log-rank test, multivariate cox regression models, and the median of overall/progression-free survival and their range values are reflected.

**Figure 3 cancers-14-02898-f003:**
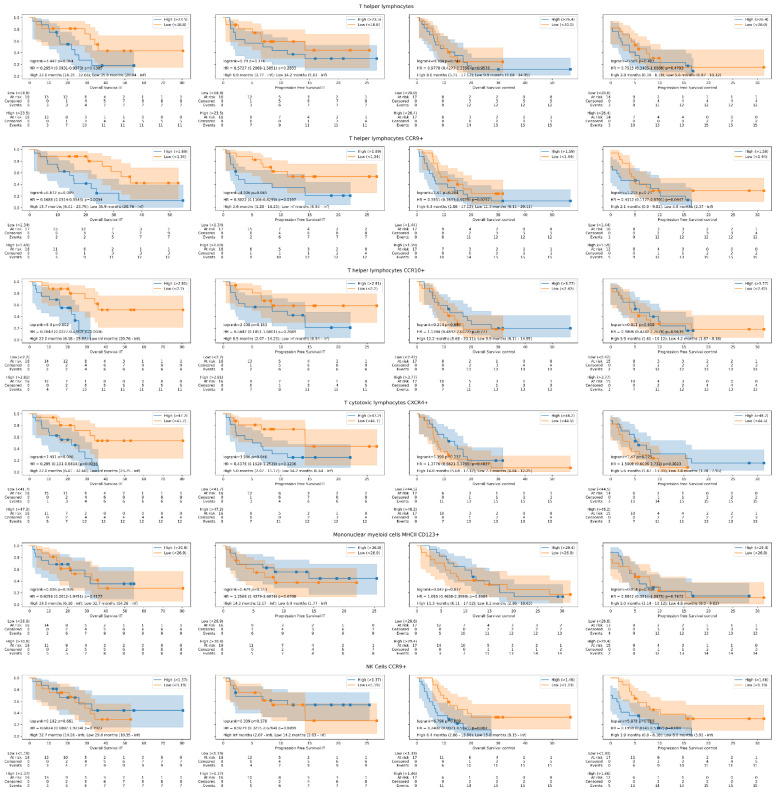
Overall survival and progression-free survival Kaplan–Meier curves in study and control groups according to the best immune biomarkers detected. The Kaplan-Meier curves show the differences in survival according the high or low expression of each peripheral blood mononuclear cell subpopulation studied in immunotherapy treated (IT) group and in control group treated with non-immunotherapy drugs. In addition, the log-rank test, multivariate cox regression models, and the median progression-free and overall survival values with their range are reflected.

**Table 1 cancers-14-02898-t001:** Immune biomarkers used for phenotypic characterization of immune cell subsets. Different subpopulations of interest were studied, up to a total of 107, through the expression or not of these basic biomarkers, making multiple combinations with them.

Immune Cell Subset	Immune Biomarkers Analyzed			
T helper lymphocytes	ADAM8	CD210	GRK2	IL6R
CD3, CD4	β7	CD47	IFNΥ	PSGL1
	CCR10	CTLA-4	IL15Ra	SLAN
	CCR9	CXCR4	IL17	Tie2
	TSP1			
T cytotoxic lymphocytes	ADAM8	CD244	IFNΥ	PD1
CD3, CD8	β7	CD47	IL15Ra	PSGL1
	CCR10	CTLA-4	IL17	SLAN
	CCR9	CXCR4	IL6R	Tie2
	CD210	GRK2	LAG3	TIM3
	TSP1			
Myeloid cells	ADAM8	CD123	GRK2	SLAN
CD14, CD11c, HLA II	β7	CD210	IL15Ra	Tie2
	CCR10	CD47	IL6R	TSP1
	CCR9	CXCR4	PSGL1	
B Lymphocytes	CD210	CD244	IL6R	
CD19	β7	CD47	PSGL1	
	CCR10	CXCR4	SLAN	
	CCR9	GRK2	Tie2	
	CD210	IL15Ra	TSP1	
Natural killer cells	ADAM8	CD244	Tie2	SLAN
CD56	β7	CD47	KIR	Tie2
	CCR10	CXCR4	NKG2A	TSP1
	CCR9	GRK2	NKG2C	
	CD210	IL15Ra	PSGL1	

**Table 2 cancers-14-02898-t002:** Clinical and demographic characteristics of the study and control groups. * Overweight definition: BMI > 25. ** Abbreviations: NSCLC: non-small cell lung cancer; BMI: body mass index; CNS: central nervous system; IrAEs: immune-related adverse events; LDH: lactate dehydrogenase.

	Immunotherapy NSCLC Cohort	Non-Immunotherapy Cohort	*p* Value
Age, median (range)	69 (50–85)	68 (43–88)	0.6
Sex			
Women	3 (7.7%)	16 (40%)	
Men	36 (92.3%)	24 (60%)	0.001
Tobacco exposure, N (%)	39 (100%)	23 (57.5%)	-
BMI, median (range)	25.12 (16.6–34.0)	23.37 (16.8–31.5)	0.4
Overweight *, N (%)	16 (41.0%)	10 (25%)	0.2
HIV, N (%)	1 (2.6%)	1 (2.5%)	1
High comorbidities (Charlson index), N (%)	7 (17.9%)	5 (12.5%)	0.49
Liver metastasis, N (%)	6 (15.3%)	22 (55%)	<0.001
CNS metastasis, N (%)	9 (23.1%)	3 (7.5%)	0.06
Previous treatments, median (range)	1 (0–3)	0 (0–2)	<0.001
Objective response, N (%)	15 (38.4%)	14 (35%)	0.8
IrAEs, N (%)	14 (35.8%)	-	-
Steroid’s consumption, N (%)	8 (20.5%)	0 (0%)	0.002
Hemoglobin, g/dL, median (range)	13.0 (7.4–17.4)	12.4 (9.1–16.3)	0.7
Neutrophils, 10^3^/mcL, median (range)	6.7 (2.4–54.0)	6.1 (2.0–15.4)	0.6
Lymphocytes, 10^3^/µL, median (range)	1.7 (0.6–5.6)	1.4 (0.3–3.8)	0.04
Platelets, 10^3^/mcL, median (range)	280.0 (135.0–721.0)	256.5 (103.0–633.0)	0.2
LDH, U/L, median (range)	201 (115–662)	167 (167–167)	<0.001

**Table 3 cancers-14-02898-t003:** Description of the best immune subpopulations biomarkers in the overall survival setting.

	NSCLC Immunotherapy Treatment Group					Non-Immunotherapy Treatment Control Group				
Biomarkers	N	Median	Range	Percentile 55 (n Patients, %)	Percentile 45 (n Patients, %)	N	Median	Range	Percentile 55 (n Patients, %)	Percentile 45 (n Patients, %)
**CD3+CD4+**	36	25.06	1.3–60.4	N = 16 23.5%	N = 16 18.8%	37	27.22	3.5–64.6	N = 17 26.4%	N = 17 20.0%
**CD3+CD4+CCR9+**	36	5.10	0.4–57.6	N = 16 1.7%	N = 17 1.3%	37	5.06	0.4–70.4	N = 17 1.6%	N = 17 1.4%
**CD3+CD4+CCR10+**	36	5.59	0.4–59.3	N = 16 2.8%	N = 16 2.2%	37	7.15	0.4–83.1	N = 17 3.7%	N = 17 2.4%
**CD3+CD8+CXCR4+**	36	50.95	27–98.1	N = 16 73.7%	N = 16 72.2%	37	48.97	27.0–98.1	N = 17 75.9%	N = 17 73.6%
**CD11c+CD14-MHCII+CD123+**	36	71.36	22.9–95.5	N = 16 79.8%	N = 16 75.8%	37	61.67	22.9–95.5	N = 17 70.7%	N = 18 68.1%
**CD56+CCR9+**	36	3.19	0.2–50.5	N = 16 1.4%	N = 16 1.2%	37	4.00	0.2–50.5	N = 17 1.5%	N = 17 1.3%

## Data Availability

Code Availability: The complete framework to run and deploy the statistical models was built using Python 3 and has been implemented on top of the lifelines, statsmodels, SciPy, and Panda libraries. The complete code for the data analysis can be accessed through this Google Colab notebook: https://colab.research.google.com/drive/1dT0MkJNXXX9t9cZE0kpdRN660h1NpShw#scrollTo=W4hDWYOZq9tQ (accessed on 8 May 2022).

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
