# Peer review of "Peripheral Blood Mononuclear Cells Predict Therapeutic Efficacy of Immunotherapy in NSCLC"

_cancers, 2022, doi:10.3390/cancers14122898_

Round 1

Reviewer 1 Report

This manuscript, entitled Peripheral blood mononuclear cells and efficacy of 
immunotherapy in patients with advanced lung  cancer, investigated biomarkers related PBMC in lung cancer. The authors collected many samples and found sound  results .

  1. the title was too big. it should be narrowed in NSCLC, and added the exact result.
  2.  the average age of subjects were more 60, did these samples have other diseases affecting results. For old  peoples usually have some basic disease such as sepsis , hypertension ..
  3.  in introduction /discussion, the authors should describe their previous work in detail and compare with others' work

Additional comments:

1.  What is the main question addressed by the research?

The manuscript  investigated evaluation of peripheral blood mononuclear

66 cells (PBMCs) subpopulations could be of value to define predictive

biomarkers of immunotherapy efficacy in advanced NSCLC  patients.

 The introduction mentioned their previous study. They should describe more in detail in introduction or in discussion part.

2. Do you consider the topic original or relevant in the field, and if

so, why?

The topic was relevant in the field for the  special issue

3. What does it add to the subject area compared with other published

material? 

The authors should discuss the previous  predictive biomarkers or diagnostic biomarkers, and compared them.

4. What specific improvements could the authors consider regarding the

methodology?

They should give statement of informed consent and protocol proved by committee. And samples treatment after collection should be more in detail.

5. Are the conclusions consistent with the evidence and arguments

presented and do they address the main question posed?

yes

6. Are the references appropriate?

 yes

7. Please include any additional comments on the tables and figures.

No

Author Response

Thanks.

Reviewer 2 Report

The authors have described the immunophenotyping characterization of subpopulations of PBMCs of patients with advance lung cancer and correlation with anti-immuno-checkpoint inhibitors treatment outcomes.

For further and detailed review, the authors must provide detailed description of the figures. Figure legends are poorly annotated. The annotated panels are lacking description and in other cases the panels are not annotated.

The authors described the analysis of 107 subpopulations however from Table I is not clear how 107 subpopulations were originated from the biomarkers used for phenotypic characterization of  immune cell subsets.

Author Response

Thanks.

Round 2

Reviewer 1 Report

The authors answered and revised all the questions in this revision .

It can be accepted.

minor :

In line 86-90, the authors  cited the same references in  three places, it was  it can be only once, or see if there were other references .

Reviewer 2 Report

The authors have extended and detailed the figure legends, though panels annotation is still missing.

Major comments

The study group population is divided into immunotherapy NSCLC cohort and non-immunotherapy cohort. While gender balance is verified in the last group, this is not the case for the first group. Comparison between similar groups could verify if the gender can explain the variance between the groups.

In Line 188 “Most frequent histology was non-squamous cell NSCLC (64.10%).  High PD-L1 expression (≥50) were presented in 56.41% of the patients.” Did the authors calculate the P-value of association between high expression and the histological type?

There is any tropism for each of the chemokine receptors. For example, the main metastization site of the two groups is different could this introduce a bias in the results.

Since there is a correlation between BMI and immunotherapy response reported previously, did the authors considered the patient variance in their analysis?

In Figure 3 the Kaplan Meyer plots are missing for the T cytotoxic lymphocytes for immunotherapy treated-patients and control groups.

Minor comments

Line 182- unfinished sentence.

Round 3

Reviewer 2 Report

Revision v3

The authors have addressed the major comments.

The authors have described the immunophenotyping characterization of subpopulations of PBMCs of patients with advance lung cancer and correlation with anti-immuno-checkpoint inhibitors treatment outcomes. The study group population is divided into immunotherapy NSCLC cohort and non-immunotherapy cohort including several tissue types. Despite the interesting approach used in the study and the relevance of assessing biomarkers in liquid biopsies for predicting treatment outcome. The study design raises some concerns. Namely, one cannot exclude the effect of chemotherapy in the distribution of subpopulations of PBMCs in the peripheral blood of cancer patients in the control cohort. On the other hand would a different type of cancer treated with ICI give a similar result, meaning are these specific for lung cancer?